# Neoadjuvant Metformin Added to Systemic Therapy Decreases the Proliferative Capacity of Residual Breast Cancer

**DOI:** 10.3390/jcm8122180

**Published:** 2019-12-11

**Authors:** Eugeni Lopez-Bonet, Maria Buxó, Elisabet Cuyàs, Sonia Pernas, Joan Dorca, Isabel Álvarez, Susana Martínez, Jose Manuel Pérez-Garcia, Norberto Batista-López, César A. Rodríguez-Sánchez, Kepa Amillano, Severina Domínguez, Maria Luque, Idoia Morilla, Agostina Stradella, Gemma Viñas, Javier Cortés, Gloria Oliveras, Cristina Meléndez, Laura Castillo, Sara Verdura, Joan Brunet, Jorge Joven, Margarita Garcia, Samiha Saidani, Begoña Martin-Castillo, Javier A. Menendez

**Affiliations:** 1Department of Anatomical Pathology, Dr. Josep Trueta Hospital of Girona, Girona 17005, Spain; elopezbonet.girona.ics@gencat.cat (E.L.-B.); goliveras@iconcologia.net (G.O.); cmelendezm.girona.ics@gencat.cat (C.M.); lcastillof.girona.ics@gencat.cat (L.C.); 2Girona Biomedical Research Institute (IDIBGI), Girona 17190, Spain; mbuxo@idibgi.org (M.B.); ecuyas@idibgi.org (E.C.); sverdura@idibgi.org (S.V.); 3Program Against Cancer Therapeutic Resistance (ProCURE), Metabolism and Cancer Group, Catalan Institute of Oncology, Girona 08908, Spain; 4Department of Medical Oncology, Breast Unit, Catalan Institute of Oncology-Hospital Universitari de Bellvitge-Bellvitge Research Institute (IDIBELL), L’Hospitalet de Llobregat, Barcelona 08908, Spain; spernas@iconcologia.net (S.P.); imorilla@iconcologia.net (I.M.); astradella@iconcologia.net (A.S.); 5Medical Oncology, Catalan Institute of Oncology, Girona 17005, Spain; jdorca@iconcologia.net (J.D.); gvinyes@iconcologia.net (G.V.); jbrunet@iconcologia.net (J.B.); 6Medical Oncology Service, Hospital Universitario Donostia, Donostia-San Sebastián 20014, Spain; isabelmanuela.alvarez@osakidetza.net; 7Biodonostia Health Research Institute, Donostia-San Sebastián 20014, Spain; 8Medical Oncology Department, Hospital de Mataró, Mataró, Barcelona 08304, Spain; smartinezpe@csdm.cat; 9IOB Institute of Oncology, Hospital Quirónsalud, Madrid & Barcelona 08023, Spain; josemanuel.perez@quironsalud.es (J.M.P.-G.); jacortes@vhio.net (J.C.); 10Medica Scientia Innovation Researcher (MedSIR), Barcelona 08007, Spain; 11Medical Oncology Service, Hospital Universitario de Canarias, La Laguna, Tenerife 38320, Spain; norberto.batista@gmail.com; 12Medical Oncology Service, Hospital Universitario de Salamanca, Salamanca 37007, Spain; rodriguez.oncologia@gmail.com; 13Instituto de Investigación Biomédica de Salamanca (IBSAL), Salamanca 37007, Spain; 14Medical Oncology, Hospital Universitari Sant Joan, Reus 43204, Spain; kamillano@grupsagessa.com; 15Medical Oncology Service, Hospital Universitario Araba, Vitoria-Gasteiz 01009, Spain; severina.dominguezfernandez@osakidetza.net; 16Department of Medical Oncology, Hospital Universitario Central de Asturias, Oviedo 33011, Spain; malucab@hotmail.com; 17Vall d’Hebron Institute of Oncology (VHIO), Barcelona 08035, Spain; 18Hereditary Cancer Programme, Catalan Institute of Oncology (ICO), Bellvitge Institute for Biomedical Research (IDIBELL), L’Hospitalet del Llobregat, Barcelona 08908, Spain; 19Hereditary Cancer Programme, Catalan Institute of Oncology (ICO), Girona Biomedical Research Institute (IDIBGI), Girona 17005, Spain; 20Unitat de Recerca Biomèdica, Hospital Universitari de Sant Joan, IISPV, Rovira i Virgili University, Reus 43204, Spain; jjoven@grupsagessa.com; 21Clinical Research Unit, Catalan Institute of Oncology, L’Hospitalet de Llobregat, Barcelona 08908, Spain; mgarciamartin@iconcologia.net; 22Unit of Clinical Research, Catalan Institute of Oncology, Girona 17005, Spain; ssaidani@idibgi.org

**Keywords:** metformin, Ki67, breast cancer, residual disease

## Abstract

The proliferative capacity of residual breast cancer (BC) disease indicates the existence of partial treatment resistance and higher probability of tumor recurrence. We explored the therapeutic potential of adding neoadjuvant metformin as an innovative strategy to decrease the proliferative potential of residual BC cells in patients failing to achieve pathological complete response (pCR) after pre-operative therapy. We performed a prospective analysis involving the intention-to-treat population of the (Metformin and Trastuzumab in Neoadjuvancy) METTEN study, a randomized multicenter phase II trial of women with primary, non-metastatic (human epidermal growth factor receptor 2) HER2-positive BC evaluating the efficacy, tolerability, and safety of oral metformin (850 mg twice-daily) for 24 weeks combined with anthracycline/taxane-based chemotherapy and trastuzumab (arm A) or equivalent regimen without metformin (arm B), before surgery. We centrally evaluated the proliferation marker Ki67 on sequential core biopsies using visual assessment (VA) and an (Food and Drug Administration) FDA-cleared automated digital image analysis (ADIA) algorithm. ADIA-based pre-operative values of high Ki67 (≥20%), but not those from VA, significantly predicted the occurrence of pCR in both arms irrespective of the hormone receptor status (*p* = 0.024 and 0.120, respectively). Changes in Ki67 in residual tumors of non-pCR patients were significantly higher in the metformin-containing arm (*p* = 0.025), with half of all patients exhibiting high Ki67 at baseline moving into the low-Ki67 (<20%) category after neoadjuvant treatment. By contrast, no statistically significant changes in Ki67 occurred in residual tumors of the control treatment arm (*p* = 0.293). There is an urgent need for innovative therapeutic strategies aiming to provide the protective effects of decreasing Ki67 after neoadjuvant treatment even if pCR is not achieved. Metformin would be evaluated as a safe candidate to decrease the aggressiveness of residual disease after neoadjuvant (pre-operative) systemic therapy of BC patients.

## 1. Introduction

Pathological complete response (pCR), the primary endpoint for a majority of neoadjuvant trials in breast cancer (BC), has been commonly adopted as a surrogate marker of long-term treatment benefit [1,2,3]. However, in patients with BC who fail to achieve pCR and have a worse prognosis, other biological markers are urgently needed to identify those at high-risk who could benefit from additional, customized therapeutic strategies. Central to these issues is Ki67, a nuclear protein associated with cellular proliferation, which is a well-established prognostic and predictive biomarker in patients with BC treated with neoadjuvant therapies. 

Pre-therapeutic Ki67 positivity is a predictive marker for pCR. Indeed, a majority of studies have identified a significant association between high Ki67-associated proliferation at baseline and higher rates of pCR after neoadjuvant chemotherapy, especially in the triple-negative and (human epidermal growth factor receptor 2) HER2-positive BC subtypes [4,5,6,7]. Although the potential prognostic value of Ki67 after neoadjuvant therapy has been less characterized, there is strong evidence to suggest that changes in Ki67 index between pre- and post-neoadjuvant hormonal and chemotherapy might be a strong predictor of outcome for patients who do not achieve a pCR. Accordingly, the evaluation not only of absolute Ki67 values, but also of any differences in specific Ki67 levels between pre- and post-neoadjuvant therapy, might predict early recurrence in BC [8,9,10,11]. Patients showing a decrease in Ki67 index from biopsy to surgery have a better prognosis than patients with high levels of Ki67 expression at surgery [8]. A difference in Ki67 expression after versus before neoadjuvant therapy might be an important predictor of early metastasis and worse outcome [10,11]. Therefore, the finding that patients with BC whose tumors have low Ki67 expression after neoadjuvant therapy show better overall and disease-free survival compared with those whose tumors maintain high Ki67 expression [8,9,11] supports the notion that patients without pCR after neoadjuvant therapy are clinically heterogeneous and could be classified according to changes in Ki67 into good and poor prognostic groups [12,13]. High post-neoadjuvant treatment-associated Ki67 index therefore indicates the need for innovative therapeutic strategies aiming to provide the protective effects of decreasing Ki67 index even if pCR is not achieved.

Here, we sought to clarify whether the anti-diabetic biguanide metformin might be employed as a safe candidate to diminish the proliferative potential of residual BC disease after neoadjuvant therapy even if pCR is not achieved. We undertook a prospective analysis involving a well-characterized intention-to-treat (ITT) cohort of the (Metformin and Trastuzumab in Neoadjuvancy) METTEN study, a randomized multicenter phase II clinical trial for patients with HER2-positive BC receiving either metformin combined with anthracycline/taxane-based chemotherapy and trastuzumab (arm A), or an equivalent regimen without metformin (arm B), before surgery [14,15,16]. We first evaluated the function of a pre-therapeutic Ki67 labeling index as a predictive marker for pCR after adding neoadjuvant metformin. We then assessed the effect of adding metformin on the change of Ki67 on sequential core biopsies in patients with residual disease after neoadjuvant treatment. Considering the open debate regarding the reproducibility of Ki67 scoring by visual analysis (VA) in multicenter settings [17,18,19,20,21], we centrally re-evaluated Ki67 using VA and an (Food and Drug Administration) FDA-cleared Ki67 automated digital image analysis (ADIA) algorithm simultaneously. Here we present preliminary evidence highlighting the clinical potential of metformin as a safe candidate to prevent and/or treat the proliferative potential of residual BC disease after neoadjuvant therapy. 

## 2. Experimental Section

### 2.1. Subjects

The METTEN study was registered with the EU Clinical Trials Register and is available online (EudraCT number 2011-000490-30). The study assessed the efficacy, tolerability, and safety of adding metformin to neoadjuvant chemotherapy plus trastuzumab in early HER2-positive BC [14]. Briefly, patients were randomly assigned to receive daily oral metformin (850 mg twice-daily) for 24 weeks concurrently with 12 cycles of weekly paclitaxel (80 mg/m^2^) plus trastuzumab (4 mg/kg loading dose followed by 2 mg/kg) followed by four cycles of 3-weekly fluorouracil (600 mg/m^2^), epirubicin (75 mg/m^2^), cyclophosphamide (600 mg/m^2^) with concomitant trastuzumab (6 mg/kg) (arm A), or equivalent sequential chemotherapy plus trastuzumab without metformin (arm B), followed by surgery. Patients had surgery within 4–5 weeks of the last cycle of neoadjuvant treatment. Post-surgery, patients received thrice-weekly trastuzumab to complete 1 year of neoadjuvant-adjuvant treatment according to institutional practice. 

The primary endpoint was the pCR rate in the per-protocol efficacy population. pCR was defined as absence of invasive tumor cells on hematoxylin and eosin evaluation of the complete resected breast specimen (and all sample regional lymph nodes if a lymphadenectomy or sentinel lymph node biopsy was performed) following the completion of neoadjuvant systemic therapy. Residual ductal carcinoma in situ only was included in the definition of pCR (ypT0/is, ypN0) [14]. Ki67 scoring was carried out in the ITT population (*n* = 79), which included all randomly assigned patients who received at least one dose of study medication. 

### 2.2. Ki67 Immunohistochemistry

Ki67 was evaluated by immunohistochemistry (IHC) on three-micron-thick sections of formalin-fixed paraffin-embedded (FFPE) tissue sections from diagnostic core and approximately one week before surgery sequential biopsies were obtained from all participating institutions of the METTEN study (Figure 1), and were subjected to Ki67 staining using the CONFIRM anti-Ki67 (30-9) rabbit monoclonal primary antibody on a Benchmark XT platform (Ventana Medical Systems Inc., Tucson, AZ, USA). 

#### 2.2.1. Visual Assessment

Ki67 was centrally evaluated by an experienced breast pathologist (ELB, Department of Anatomical Pathology, Dr. Josep Trueta Hospital of Girona, Girona, Spain), blinded to both the original pathology reports and the METTEN clinical trial database throughout the entire procedure. Manual counting of negative and positive cells was carried out by “eyeballed” estimates involving a more global average and not only hot spots. 

#### 2.2.2. Automated Digital Image Analysis

An FDA-cleared Ki67 image analysis algorithm was employed for determining Ki67. First, all of the stained slides were scanned into digital images using the VENTANA iScan System Version 1.0 (Ventana Medical Systems, Inc., Sunnyvale, CA, USA). Then, the VENTANA Companion Algorithm Ki67 (30-9) image analysis application was employed with the VENTANA iScan Coreo Au scanner running the Virtuoso Digital Pathology Image Analysis software. The same breast pathologist who evaluated Ki67 scores by VA selected the regions of interest for ADIA-based Ki67 scoring, taking into consideration confounding factors of digital scoring such as intratumor heterogeneity, presence of intermixed inflammatory cells, carcinoma in situ, or the occurrence of other non-cancer proliferating cells. 

### 2.3. Statistical Analysis

Descriptive data were summarized using percentages, medians or means with their respective 25th and 75th percentiles, or standard deviations, as appropriate. Clinical baseline characteristics between treatment arms were assessed using the Chi-square or Fisher’s exact test for categorical variables, Student’s t-test for continuous variables with normal distribution, or the Mann–Whitney U test for non-normal distributions. The assumption of normality was evaluated with the Shapiro–Wilk test in parametric testing. 

Agreement between VA and ADIA of Ki67 scoring was assessed using the Bland–Altman plot with 95% limits of agreement (LOA). Bland–Altman plots were constructed to enable visual observation for agreement between the two methods and to determine the 95% LOA. A non-parametric Passing–Bablok regression analysis, in which the slope of the regression line is calculated as a shifted median of all possible slopes between pairs of points that is resistant to outliers thereby giving an unbiased slope estimate solely if both methods have constant coefficients of variation, was applied to determine the interchangeability between the VA- and the ADIA-based Ki67 scoring by assessing systematic error: (a) constant bias was confirmed when the 95% confidence interval (CI) for intercept included value “0” and (b) proportional bias when 95% CI for slope included value “1.” Intra-class correlation coefficient (ICC) with a 95% CI was estimated using two-way mixed effect models to assess the consistency between VA and ADIA assessment of Ki67 scoring according to the two score methods. A higher ICC indicates better consistency. Although there is no universally accepted, standardized criteria for the ICC, we used the following kappa coefficient-like criteria to aid interpretation (20): 0.00–0.20 was interpreted as slight correlation; 0.21–0.40 as fair correlation; 0.41–0.60 as moderate correlation; 0.61–0.80 as substantial correlation; and >0.80 as excellent (almost perfect) correlation. 

Binary logistic regression was used to assess the prognostic effect of Ki67 scoring on pCR. Unadjusted and adjusted odds ratios (ORs) with their relative 95% CIs were reported as a measure of association. All tests were two-sided and a *p*-value < 0.05 was set as statistically significant. Statistical analyses were carried out using SPSS (IBM Corp. released 2017. IBM SPSS Statistics for Windows, Version 25.0; Armonk, NY, USA) and STATA (StataCorp. 2013. Stata Statistical Software: Release 13; StataCorp LP, College Station, TX, USA).

### 2.4. Ethics Statement 

The hospital (Dr. Josep Trueta Hospital, Girona, Spain) ethics committee (Clinical Investigation Ethic Committee, CIEC) and independent institutional review boards at each site participating in the METTEN study approved the protocol and any amendments. All procedures were in accordance with the ethical standards of the institutional research committees and with the 1964 Helsinki declaration and its later amendments or comparable ethical standards. Informed consent was obtained from all individual participants included in the study. 

## 3. Results

### 3.1. Study Participants

The present study was designed to evaluate the therapeutic activity of neoadjuvant metformin with respect to Ki67 on sequential core biopsies obtained from patients belonging to the ITT population of the METTEN trial (Figure 1), which included all randomly assigned HER2-positive BC patients who received at least one dose of study medication (*n* = 79; Table 1). The baseline characteristics and clinical-pathological variables at diagnosis of those ITT patients who achieved pCR after neoadjuvant therapy and those who did not have been reported previously [15,16]. 

### 3.2. Correlation between Visual Assessment and Automated Digital Image Analysis for Ki67 Scoring

Of the 84 tissue samples included in the VA-ADIA correlation study (*n* = 69 from available diagnostic core biopsies and *n* = 15 from available sequential biopsies obtained following preoperative therapy and approximately one week before surgery), the mean VA Ki67 score from an experienced breast pathologist was 40.36% (95% CI: 35.76%–44.95%), whereas the mean Ki67 score by ADIA was 39.44% (95% CI: 34.39%–44.49%) (Figure 2A). VA and ADIA scoring of Ki67 was highly correlated (Spearman’s correlation coefficient 0.839 (*p* < 0.001)). Passing–Bablok regression analysis showed a strong linear relationship between VA and ADIA methods (Pearson’s correlation coefficient 0.852 (*p* < 0.001)), with a constant (intercept −3.35; 95% CI −10.0–0.26) and non-proportional bias (slope 1.08; 95% CI 0.97–1.22) (Figure 2B). The Bland–Altman plot (difference plot) with 95% LOA showed a small and balanced spread of the relative difference between VA and ADIA, with an estimated bias (mean difference) of −0.92 (95% LOA −24.95 to −23.11). Only 3 out of 84 values (3.6%) felt outside the LOA (Figure 2C). 

To evaluate whether there was non-uniformity within specific data ranges, the consistency between VA and ADIA was analyzed using the ICC. We selected the two-way mixed-effects model to compute both the absolute agreement (AA, when different raters assign the same score to the same subject) and the consistency (C, when raters’ scores to the same group of subjects are correlated in an additive manner) definitions of ICC. By ICC analysis, a good agreement was demonstrated between VA and ADIA Ki67 scoring in the whole cohort of our study (ICC = 0.849 (AA)/0.848 (C), 95% CI: 0.777–0.899, *p* < 0.001). Because Ki67 cut-offs ranging from 10% to 30% have been widely employed when classifying patients into Ki67 high- or low-risk groups for clinical decision making, we re-classified Ki67 scoring into three groups (≤10%, 11%–30% and >30% Ki67), stratified by VA values, according to the two score methods (Figure 2D). The 11%–30% and ≤10% (ICC = 0.281 (AA)/277 (C) and 0.117/0.114, *p* = 0.100 and 0.386, respectively; Table 2) groups showed notably poorer consistency than the >30% group, which reached a substantial correlation (ICC = 0.735 (AA)/0.739 (C), *p* < 0.001; Table 2). When Ki67 scoring was stratified into two groups (low 0%–30%; high >30%), we concluded the level of reliability to be moderate for low Ki67 values (ICI = 0.563 (AA)/0.565 (C), *p* < 0.001) in comparison with the substantial level of correlation observed for high Ki67 values. 

### 3.3. Association between VA- and ADIA-Based Ki67 Scoring and Pathological Response

Because the clinically relevant Ki67 cut-off is 20% as defined by the St. Gallen criteria [22], we applied this cut-off to investigate the association between Ki67 expression, treatment arm, and pCR (Appendix A). In bivariate analysis, we observed the predictive capacity of baseline ADIA-based Ki67 ≥20% to significantly associate with the probability of achieving pCR (OR = 4.76, 95% CI = 1.13–20.09, *p* = 0.034; Figure 3). The direction and/or intensity of the predictive relationship between baseline ADIA-based Ki67 ≥20% and pCR occurred independently of the treatment arm (Appendix A). After additional adjustment for potential confounding characteristics such as the hormone receptor status, the relationship between baseline ADIA-based Ki67 ≥20% and the ability of neoadjuvant trastuzumab-based chemotherapy to achieve a pCR in patients remained significant (OR = 5.57, 95% CI = 1.26–24.74, *p* = 0.024; Figure 3; Appendix A). We failed to observe such predictive capacities of Ki67 ≥20% when Ki67 scoring was stratified using the VA values (Figure 3; Appendix A). 

### 3.4. Impact of Neoadjuvant Metformin on Ki67 Expression in Non-pCR Patients

We analyzed the decrease (or lack of decrease) in the percentage of Ki67-positive cancer cells between paired samples of core biopsies samples at baseline and approximately one week before surgery obtained from non-pCR patients in whom both VA and ADIA-based Ki67 scoring was available. The median baseline Ki67 level in this population (*n* = 14) was 42% (VA)/34.5% (ADIA) before neoadjuvant trastuzumab-based chemotherapy, which significantly decreased to a median of 19.5% (VA; *p* = 0.014)/17.0% (ADIA; *p* = 0.009) after treatment (Figure 4). To determine whether the addition of metformin significantly impacted Ki67 expression during neoadjuvant trastuzumab-based chemotherapy, we re-evaluated the relative change in Ki67 scoring from baseline to end-point treatment in non-pCR patients (Figure 4). The Ki67 decreases were largest in the metformin-containing arm A (*n* = 8), from 42.5% (VA)/34.5% (ADIA) at baseline to 11% (VA; *p* = 0.025)/11% (ADIA; *p* = 0.035) at surgery (Figure 4). Conversely, no significant differences were observed in the control treatment arm B (*n* = 6), in which Ki67 decreased from 36.0% (VA)/38.5% (ADIA) at baseline to 28.5% (VA; *p* = 0.293)/25.0% (ADIA; *p* = 0.080). 

Finally, we evaluated the change of Ki67 category (high ≥20% versus low <20%) by treatment arm, finding that 57% (VA)/50% (ADIA) of high-Ki67, non-pCR patients moved into the low-Ki67 category following neoadjuvant treatment in the metformin-containing arm A (Figure 5). Conversely, 67% (VA)/80% (ADIA) of high-ki67, non-pCR patients remained unchanged following neoadjuvant treatment in the control arm B (Figure 5). This numerically higher pre- and post-treatment change of Ki67 scoring categories in the metformin-containing arm A was not statistically significant compared with that observed in the standard treatment arm B (*p* = 0.5210 VA; *p* = 0.3582 ADIA).

## 4. Discussion

The prognosis of patients with BC is better for those who obtain pCR with neoadjuvant therapy than for those who do not [23,24]. For non-pCR patients—a clinically heterogenous population in terms of prognosis—there are few available therapeutic strategies (i.e., capecitabine in HER2-negative BC [25] and trastuzumab emtansine in HER2-positive BC [26]) that can be added to neoadjuvant therapy regimens to prevent recurrence. Ki67 is a validated biomarker of recurrence-free survival in BC residual disease after neoadjuvant therapy [27]. To the best of our knowledge, this is the first study reporting that the addition of neoadjuvant metformin to a well-established pre-operative treatment involving chemotherapy and targeted therapy significantly impacts the short-term changes in Ki67 post-neoadjuvant therapy. 

Metformin, a biguanide derivative that has long been a cornerstone in the treatment of type 2 diabetes, has recently become incorporated into the armamentarium against cancer. Epidemiological and preclinical evidence is beginning to suggest that metformin may reduce overall cancer risk and mortality, with particularly significant effects in BC [28,29,30,31,32,33,34,35,36,37]. Accordingly, many clinical studies, including proof-of-principle studies in the prevention setting and phase 2 trials in the adjuvant and metastatic settings, have been planned and/or are currently under way to test the causal nature of the suggested correlation between metformin and clinical benefit in cancer. Among them, several pre-operative window-of-opportunity trials have consistently demonstrated the ability of metformin, at conventional anti-diabetic doses, to reduce Ki67 expression in non-diabetic cancer patients, which might vary with host and tumor characteristics [31,38,39,40,41]. Meta-analyses of randomized clinical trials investigating the effect of metformin on biomarkers associated with BC outcomes have suggested a potential benefit from metformin treatment in reducing Ki67 expression in these patients [42,43]. However, no study to date had explored whether the addition of metformin to neoadjuvant regimens in BC might impact the Ki67-measured proliferative activity of residual tumors when pCR is not achieved. Because the quality of evidence regarding the potential benefit of metformin treatment in reducing Ki67 proliferation rates has recently been questioned due to inadequate methodology [44], and considering the open debate regarding the reproducibility of Ki67 technical assessment, interpretation, and scoring in multicenter settings [17,18,19,20,21], we decided to centrally re-evaluate Ki67 using both VA and ADIA approaches simultaneously. Because the consistency over time of paired, sequential biomarker measurements (e.g., Ki67) can be affected not only by intra-tumoral heterogeneity but also by the biospecimen type [45], we employed sequential core biopsies (but not core versus resection), which have been shown to be more consistent and appropriate to assess the effects of drug therapy in vivo on Ki67 using immunohistochemistry [46]. Such an augmentation of the level of evidence suggests that metformin’s ability to reduce the post-neoadjuvant treatment Ki67 index is a real biological phenomenon rather than an analytical artifact or a tissue sampling bias. Although the mean Ki67 suppression was similar between the two treatment arms, a statistically larger decrease in the proliferative capacity of residual tumor cells from baseline values was observed only in the non-pCR patients belonging to the metformin-containing arm, thereby suggesting that metformin might be considered as a safe candidate to prevent and/or treat the proliferative potential of residual BC disease after neoadjuvant therapy. 

Neoadjuvant therapy is known to affect the Ki67 index based on the ability of therapeutic agents to act either on cycling cells (e.g., cytotoxic chemotherapeutics) or on major oncogene-driven proliferative pathways (e.g., HER2-targeted drugs). The numerically higher pCR rate observed in the METTEN patients receiving neoadjuvant metformin compared with the reference arm did not reach statistical significance [14]. However, adding neodjuvant metformin appears to provide the protective effects of decreasing Ki67 in the residual tissue that remains after treatment with conventional anthracycline/taxane cytotoxic chemotherapy plus anti-HER2 trastuzumab neoadjuvant therapy. These findings, overall, might suggest that the ability of metformin to target the relationship between lowered Ki67 and inhibition of tumor growth could occur in the absence of objective tumor regression and likely involves pathobiological features of post-therapy residual BC disease that are different to treatment-naïve BC. Beyond the intrinsic (genomic) and/or acquired (nongenomic) mechanisms driving the survival of residual tumors after neoadjuvant therapy, we are accumulating evidence that the ability of cancer cells to flexibly rewire their mitochondrial metabolism (e.g., enhanced oxidative phosphorylation, lipid metabolism, and serine/glycine one-carbon metabolism) critically contribute to therapy resistance in residual disease [47,48,49]. Because all these mitochondria-centric traits can be targeted by metformin [50,51,52,53], it is tempting to speculate that adding metformin to complex neoadjuvant regimens involving chemotherapy and targeted therapy might impede the activation of compensatory metabolic programs linked to the maintenance of proliferative capacity in residual disease [47,48,49]. Nonetheless, because the inhibition of Ki67 after neoadjuvant therapy has a prognostic role [27], it would be of interest to test whether this greater proliferation inhibition that resulted in half of all patients exhibiting high-Ki67 at baseline moving into the low-Ki67 (<20%) category at surgery will be influential in recurrence-free survival among patients not achieving pCR. If future studies will conclude that the neoadjuvant addition of metformin is effective in prolonging disease-free survival among the patients who had residual invasive disease on pathological testing, metformin should become a safe candidate to decrease the aggressiveness of residual disease in the preoperative systemic treatment of breast cancer patients. Evaluation of safety and tolerability of the six-month intervention with metformin as part of a complex neoadjuvant combination including chemotherapy and trastuzumab showed no difference with that of the equivalent regimen without metformin [14]. The most common adverse effects (AEs) of grade ≥3 were neutropenia in both arms and diarrhea in the metformin-containing arm, while none of the serious AEs was deemed to be metformin-related [14]. Although a major concern regarding the clinical usage of metformin is its known ability to induce gastrointestinal upset and diarrhea, which might limit patient compliance when combined with cytotoxic chemotherapy [54], it should be noted that the dropout rate in the metformin arm was much lower than the expected (25%)—solely 13% of patients withdrew because of metformin-related upset and diarrhea, whereas 75% of patients completed the triple regimen of metformin, chemotherapy, and trastuzumab [14]. 

## 5. Limitations of the Study

Our current findings must be interpreted with caution and at least two major limitations should be borne in mind. First, we acknowledge that our conclusions would be at best hypothesis generating given both the post-hoc nature of the study for which the original design was not powered and the small sample size evaluated. Second, both the variation in the Ki67 scoring system between VA and ADIA and the lack of Ki67 scores of <14% on diagnostic core biopsies precluded further analyses using recommendations from the International Ki67 in Breast Cancer working group [55,56].

## 6. Conclusions

Classic histopathological parameters such as Ki67 status provide valuable prognostic and predictive information when assessed in residual BC tissue after neoadjuvant therapy. The proliferative capacity of residual BC disease indicates the existence of partial treatment resistance and higher probability of tumor recurrence, which is supported by the negative prognostic value associated with the persistence of tumor cells following neoadjuvant therapies [12,27,57,58,59]. Because the decision-making process for the choice of adjuvant treatment in BC is rapidly transitioning from the use of therapeutic approaches based on the assessment of pre-neoadjuvant therapy features to being guided by the parameters of the residual disease after neoadjuvant therapy, further research is warranted on the potential of metformin as a preemptive treatment of residual BC proliferation. 

## Figures and Tables

**Figure 1 jcm-08-02180-f001:**
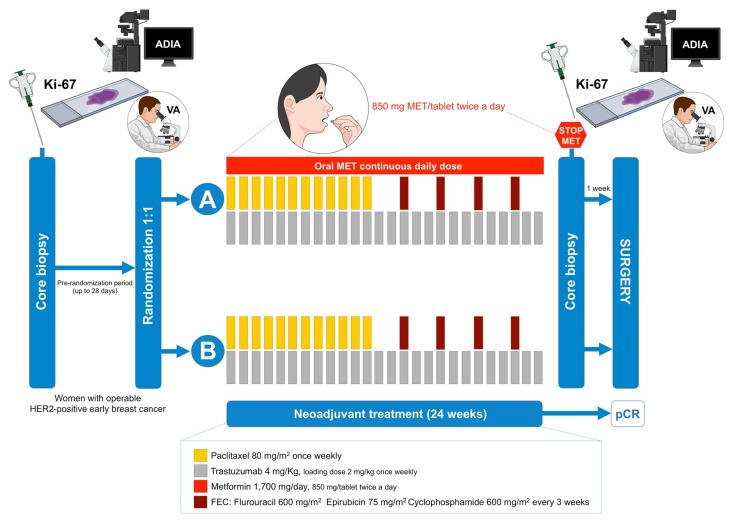
METTEN study design. Figure shows the treatment and sequential biopsies schedules of the METTEN study (MET: metformin; pCR: pathological complete response; ADIA: automated digital image analysis; VA: visual analysis).

**Figure 2 jcm-08-02180-f002:**
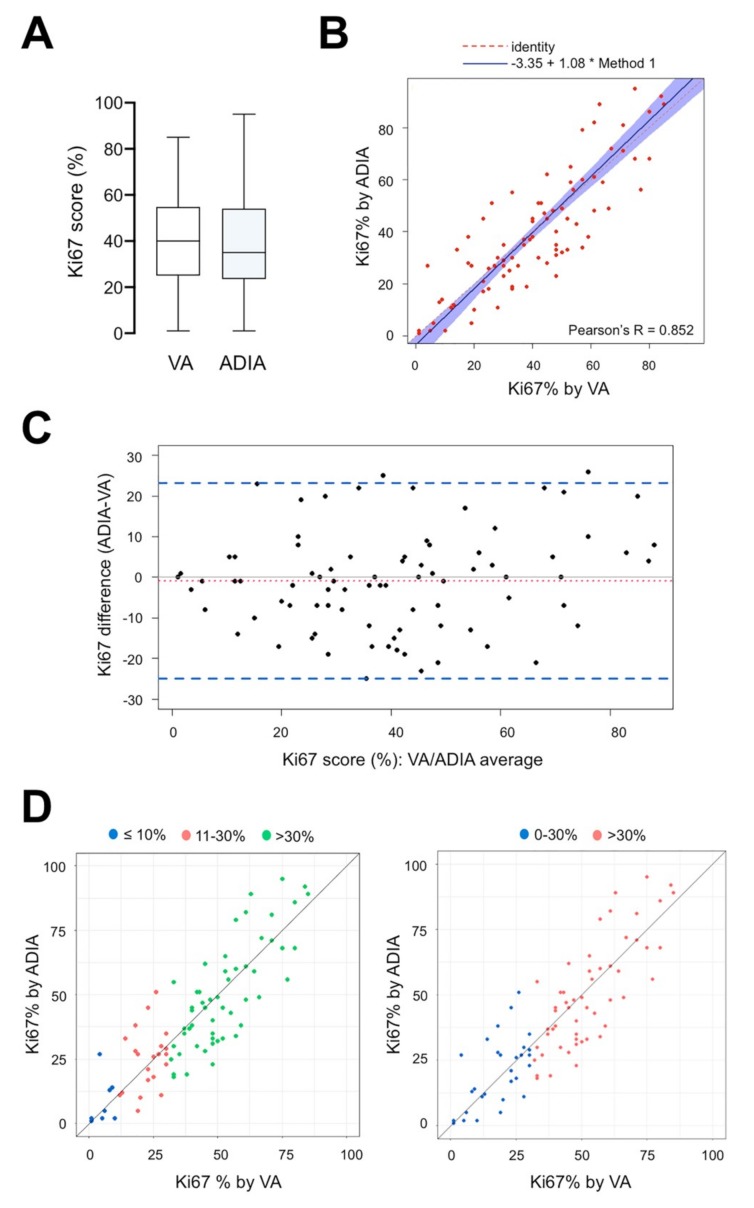
Performance fit comparison between manual visual assessment and automated digital image analysis of Ki67 score on diagnostic and pre-operative (approx. 1-week before surgery) core biopsies in the METTEN study (*n* = 84). (**A**) Box plots of Ki67 scores (%) according to visual assessment (VA) and automated digital image analysis (ADIA) methods. Horizontal lines inside the boxes represent the median value; box limits indicate the 25th and 75th percentiles; whiskers extend 1.5 times the interquartile range from the 25th and 75th percentiles. (**B**) Scatter plot of the Passing–Bablok regression analysis of paired Ki67 scores obtained by VA and ADIA. Blue-shaded areas represent 95% confidence intervals. R^2^ value and regression equation are indicated in the figure. (**C**) Limits of agreement (LOA). The difference in Ki67 measurements made in paired analysis was plotted against the average of two methods compared. Figure depicts the Bland–Altman scatter plot of agreement between Ki67 scores obtained by VA and ADIA. General agreement between methods is evident with minimal variance (LOA, area between dashed lines). (**D**) Correlation between VA and ADIA of Ki67 scores stratified by VA values.

**Figure 3 jcm-08-02180-f003:**
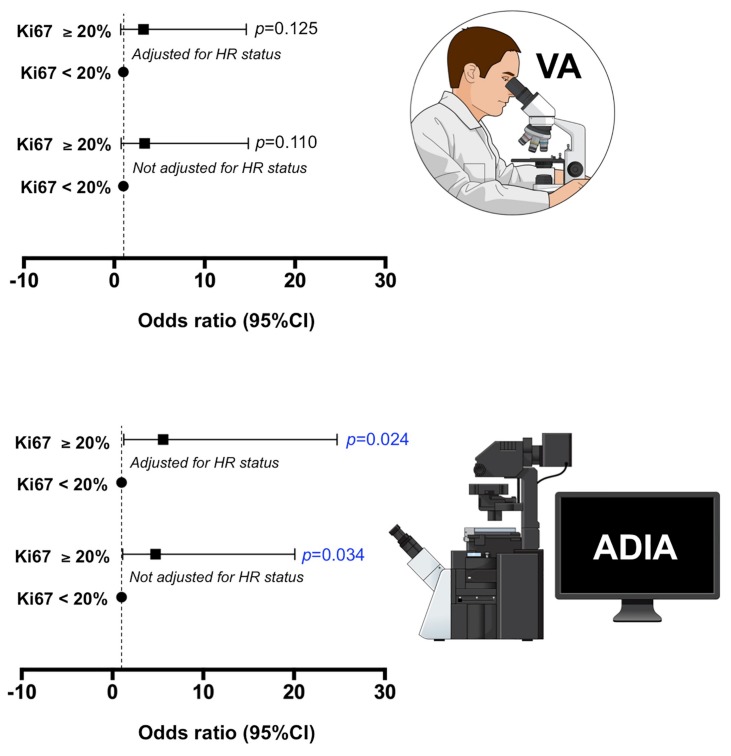
Association of baseline Ki67 scores obtained by visual assessment and automated digital image analysis and the ability of neoadjuvant treatment to achieve pathological complete response (pCR) at surgery in the METTEN study. We applied the clinically relevant Ki67 cut-off of 20% (St. Gallen criteria) to investigate the association between low (<20%) and high (≥20%) Ki67 expression at baseline and the ability of neoadjuvant treatment to achieve pCR at surgery. The association was further adjusted by a well-known predictive factor of pCR in the neoadjuvant treatment of breast cancer, such as the hormonal receptor status.

**Figure 4 jcm-08-02180-f004:**
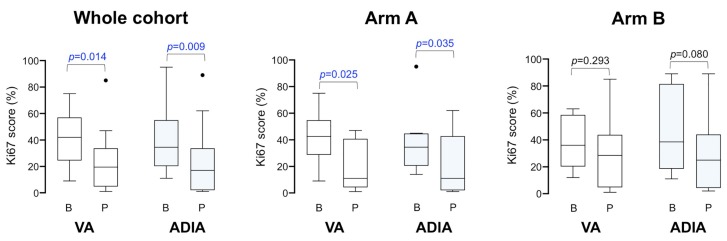
Global changes in Ki67 scores among METTEN study patients with a residual tumor after neoadjuvant treatment. Box plots showing the distribution of Ki67 values in baseline (B) and pre-operative (P) core biopsies in the whole population and stratified by treatment arms. The figure shows the median values (horizontal bars within boxes) and 25th and 75th percentile (lower and upper horizontal lines of the boxes); whiskers extend 1.5 times the interquartile range from the 25th and 75th percentiles. Circles: outliers.

**Figure 5 jcm-08-02180-f005:**
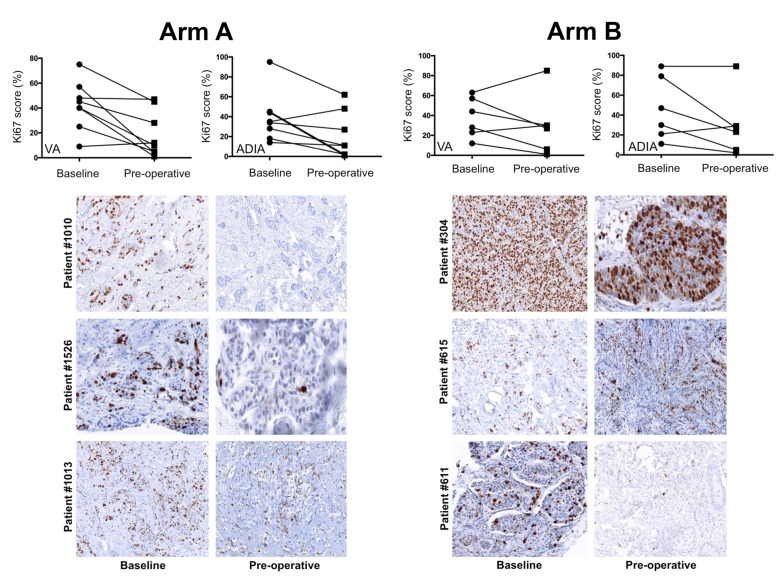
Individual changes in Ki67 scores among METTEN study patients with a residual tumor after neoadjuvant treatment. Top: changes in Ki67 score for individual patients at baseline and post-treatment according to treatment arm and scoring method. Bottom: representative images of Ki67 expression staining are shown for each pair of biopsies at baseline and approximately one week before surgery (pre-operative) in individual patients.

**Table 1 jcm-08-02180-t001:** Characteristics of patients at baseline according to treatment arm.

	Arm A (*n* = 38)	Arm B (*n* = 41)	*p*-Value
**Age** median (p25; p75)	47 (39.5; 54.3)	47 (40.5; 57)	
**pCR**			
No	17 (44.7%)	14 (34.1%)	0.335
Yes	21 (55.3%)	27 (65.9%)	
**HR status**			
ER and/or PR+	19 (50.0%)	24 (58.5%)	0.447
ER and PR−	19 (50.0%)	17 (41.5%)	
**Tumor tissue** ^a^			
No	4 (10.5%)	4 (9.8%)	1.000
Yes	34 (89.5%)	37 (90.2%)	
**Ki67 VA (%)** ^b^			
Mean ± SD	43.74 ± 20.40	43.67 ± 18.11	0.988
Median (p25; p75)	41 (29; 58)	46.5 (28.5; 68.2)	
(min; max)	(8; 80)	(12; 84)	
**Ki67 VA (%)**			
<20%	4 (11.8%)	5 (13.9%)	1.000
≥20%	30 (88.2%)	31 (86.1%)	
**Ki67 ADIA (%)** ^b^			
Mean ± SD	42.41 ± 21.34	42.56 ± 22.12	0.978
Median (p25; p75)	40 (27.25; 52.25)	38.0 (27.0; 58.25)	
(min; max)	(10; 95)	(5; 92)	
**Ki67 ADIA (%)**			
<20%	7 (20.6%)	4 (11.6%)	0.276
≥20%	27 (79.4%)	32 (88.9%)	

pCR, pathological complete response; HR, hormone receptor; VA, visual assessment; ADIA, automated digital image analysis; ER, estrogen receptor; PR, progesterone receptor; ^a^ missing cases (*n* = 4, arm A; *n* = 4, arm B); ^b^ missing cases (*n* = 4, arm A; *n* = 5, arm B).

**Table 2 jcm-08-02180-t002:** Intra-class correlation coefficient between visual assessment and automated digital image analysis of Ki67 scoring.

Group	*n*	ICC (95% Confidence Interval) Absolute Agreement	ICC (95% Confidence Interval) Consistency
Whole cohort	84	0.849 (0.777–0.899) *p* < 0.001	0.848 (0.775–0.899) *p* < 0.001
0%–30%	30	0.563 (0.265–0.764) *p* < 0.001	0.565 (0.262–0.766) *p* < 0.001
≤10%	8	0.117 (−0.625–0.731) *p* = 0.386	0.114 (−0.598–0.725) *p* = 0.386
11%–30%	22	0.281 (−0.153–0.623) *p* = 0.100	0.277 (−0.154–0.619) *p* = 0.100
>30%	54	0.735 (0.584–0.837) *p* < 0.001	0.739 (0.589–0.840) *p* < 0.001

ICC, intra-class correlation coefficient.

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
