# Peer review of "Neoadjuvant Metformin Added to Systemic Therapy Decreases the Proliferative Capacity of Residual Breast Cancer"

_jcm, 2019, doi:10.3390/jcm8122180_

Round 1
Reviewer 1 Report
I have the following concerns:
1. There is no mentioning of the no. of patients in each arm, which immediately raises concern on the power of the study.
2. We know from previous literature that comparing sequential core biopsies, but not core versus surgical specimen is the most consistent and appropriate way to assess the effect of drug therapy in vivo. Published research on the concordance between diagnostic core biopsies and surgical specimen for Ki67 ranges between 58% to 82% [i].
3. Intra-tumoral heterogeneity ; variation in the scoring system between Visual Assessment (VA) and Automated Digital Image Analysis (ADIA); using a cut-off point of 20% (and not 13.25% in concordance with expert opinion [ii]), are all potential biases in this paper.
4. Between the two arms, the number of patients with Ki67 score of <20% was 9 (using VA system), and 11 (using the ADIA system). This puts the power of the study under question.
References:
[i] BMC Cancer. 2016 Sep 22;16(1):745.
A prospective comparison of ER, PR, Ki67 and gene expression in paired sequential core biopsies of primary, untreated breast cancer.
Hadad SM1, Jordan LB2, Roy PG3, Purdie CA2, Iwamoto T4, Pusztai L5, Moulder-Thompson SL6, Thompson AM7.
[ii] J Natl Cancer Inst. 2011 Nov 16;103(22):1656-64. doi: 10.1093/jnci/djr393. Epub 2011 Sep 29.
Assessment of Ki67 in breast cancer: recommendations from the International Ki67 in Breast Cancer working group.
Dowsett M1, Nielsen TO, A'Hern R, Bartlett J, Coombes RC, Cuzick J, Ellis M, Henry NL, Hugh JC, Lively T, McShane L, Paik S, Penault-Llorca F, Prudkin L, Regan M, Salter J, Sotiriou C, Smith IE, Viale G, Zujewski JA, Hayes DF; International Ki-67 in Breast Cancer Working Group.
Reviewer 2 Report
The authors present an analysis of their METTEN study based on the evaluation of the Ki67 index in the ITT population. The article is clearly written and the statistics which support the correlation of Ki67 index with various observations are described in details.
The authors recognize that the METTEN study was underpowered and thus, the results this study suffer from the same weakness. Statistics should thus be interpreted with caution.
Since methods to determine a Ki67 score are subject to debate, they compare two methods to measure Ki67 (visual VA vs automated ADIA). It is written that assumption of normality was done but does the differences between VA and ADIA measurement follow gaussian distribution and what is the Shapiro-Wilk value?
They evaluate the association of baseline Ki67 index > 20% and pCR. Only ADIA-based Ki67 is correlated to pCR achievement. However the authors show in section 3.2 that the 11-30% zone of the Ki67 score has the less correlation between the two methods. Therefore, is the apparent success of ADIA-based prediction significant?
Minor point: in abstract and in conclusion, line 41 and 412, it is written "proliferative capacity of residual breast cancer is a molecular substrate". The meaning of molecular substrate is not very clear in this context.
Reviewer 3 Report
The present manuscript reports the therapeutic activity of neoadjuvant metformin with respect to Ki67. Authors have conducted the study on patients belonging to ITT population. This paper has a potential to be accepted, but following are some minor issues associated with publication:
What was the route of drug administration, age and sex of patients? Authors should mention these in methods. Did authors check for the adequate organ function (based on platelet, neutrophil count, AST, ALT, creatinine, serum total bilirubin, etc)? Was neoadjuvant metformin therapy associated with any adverse events such as bone marrow toxicity?Author Response
Please see the attachment

Round 2
Reviewer 1 Report
n/a